# A quantitative analysis of semantic information in deep representations of text and images

**Santiago Acevedo**[*]                                                                  *sacevedo@sissa.it*
*Scuola Internazionale Superiore di Studi Avanzati (SISSA)*

**Andrea Mascaretti**[*]                                                                *amascare@sissa.it*
*Scuola Internazionale Superiore di Studi Avanzati (SISSA)*

**Riccardo Rende**                                                                        *rrende@sissa.it*
*Scuola Internazionale Superiore di Studi Avanzati (SISSA)*

**Matéo Mahaut**                                                                  *mateo.mahaut@upf.edu*
*Universitat Pompeu Fabra (UPF)*

**Marco Baroni**                                                                  *marco.baroni@upf.edu*
*Catalan Institute of Research and Advanced Studies (ICREA) and Universitat Pompeu Fabra (UPF)*

**Alessandro Laio**                                                                           *laio@sissa.it*
*Scuola Internazionale Superiore di Studi Avanzati (SISSA)*

**Reviewed on OpenReview:** *https://openreview.net/forum?id=sBnaFSIuGR*

## Abstract

It was recently observed that the representations of different models that process identical or semantically related inputs tend to align. We analyze this phenomenon using the Information Imbalance, an asymmetric rank-based measure that quantifies the capability of a representation to predict another, providing a proxy of the cross-entropy which can be computed efficiently in high-dimensional spaces. By measuring the Information Imbalance between representations generated by DeepSeek-V3 processing translations, we find that semantic information is spread across many tokens, and that semantic predictability is strongest in a set of central layers of the network, robust across six language pairs. Within this pool of languages, we find that English representations tend to be more predictive than the others. Restricting the analysis to English inputs, we observe that DeepSeek-V3 representations are more predictive of those produced by a smaller model such as Llama3-8b than the opposite. In the visual domain, we observe that semantic predictability concentrates in middle layers for autoregressive models and in final layers for encoder models, and these same layers yield the strongest cross-modal predictability with textual representations of image captions. Our results support the hypothesis of semantic convergence across languages, modalities, and architectures, while showing that directed predictability between representations varies strongly with layer-depth, model scale, and language.

## 1 Introduction

Large transformers encode information in high-dimensional spaces, transforming representations across layers to perform a task. The Platonic Representation Hypothesis (Huh et al., 2024) suggests that–at large model sizes–representations of semantically related inputs converge to similar neighboring structures regardless of (i) the task of the model and (ii) the specific encoding of the information. The statistical bias driving the convergence of neighborhood is, plausibly, the *meaning* which is shared between related inputs, with models

loosely acting as mappings from data-specific representations to a hidden manifold, shared between different data modalities, in which semantically similar concepts are close to each other.

One can cast the convergence to a universal representation as a mutual information problem. If one models the representations of related inputs, say an image and its caption, as random variables $x$ and $y$ with a joint probability distribution $p(x, y)$, one can measure the information lost by replacing the joint distribution with the product of the marginals $p(x)p(y)$. If the distributions are independent, observing the caption is not informative of the image. If representations converge, a caption will be highly informative about the respective image, and the joint distribution is not approximated well by the marginals. Mutual information alone, however, is not sufficient to compare different models. In the deep layers, the distributions $p(x)$ and $p(y)$ loosely span a common underlying manifold, which is hypothesized to be associated with shared semantic information. The support of the different distributions does not reduce to this manifold, but spans other directions, orthogonal to the shared manifold, which are data- and architecture-specific. To quantify the relative information of $x$ and $y$ we need a statistical measure that is asymmetric, as there may be a partial order relation between models and representations induced by model size, model quality, different data-specific subspaces, etc. The cross-entropy is a measure of how difficult it is to encode an event from a distribution into another distribution, and it would be the ideal choice, but estimating it is computationally difficult due to the dimension of the representations, which are of order of thousands. To overcome this limitation, we employ here the Information Imbalance (Glielmo et al., 2022), a statistical measure that can be linked to a conditional cross-entropy between the ranks (see Appendix A), and can be computed efficiently even in high-dimensional spaces. This statistics has already been successfully used to perform feature selection(Wild et al., 2025) and to detect the presence of causal links (Del Tatto et al., 2024; Allione et al., 2025). The Information Imbalance between representation $x$ and representation $y$ is proportional to the average rank in representation $y$ of the the data points which are nearest neighbors in representation $x$. If representation $x$ predicts representation $y$, this average will be small, since the nearest neighbors of in $x$ are also nearest neighbors $y$. If, instead, $x$ does not predict $y$ well, the average rank will be high, since the nearest neighbors in $x$ are not necessarily close also in $y$. Note that the measure is asymmetric, as the nearest neighbors in one space could be near neighbors in the other space even if the reverse does not hold (Glielmo et al., 2022). A simple example of two variables being asymmetrically informative about each other is that of a parabola, $y = x^2$, in which $x$ is more informative, since knowing $x$ allows predicting $y$, whereas the opposite is not true. For a detailed illustration of the II in this case and others, please see Glielmo et al. (2022). The importance of asymmetric metrics when comparing representations was already stressed in the model stitching literature (Bansal et al., 2021), and it is going to be further illustrated in the rest of this paper.

In this work, we first show on synthetic datasets that the Information Imbalance can be used to assess quantitatively the information provided by a representation on another representation, and we compare its properties to those of other statistics used in the literature for similar purposes. We then focus on DeepSeek-V3, the largest publicly available LLM, and measure, layer by layer, the relative information content between representations of sentence pairs that are translations of each other. This reveals a region of the network, robust across language pairs, that consistently encodes shared semantic content. We find clear information asymmetry effects, such that English-sentence representations tend to be better predictors of their translation than the other way around, and DeepSeek-V3 better predicts the smaller Llama3 models than the reverse. We also compare different methods for representing a sentence, considering the concatenation of the token activations, the mean of token activations, and the last token alone. We find that the convergence of representations is better revealed by utilizing several tokens, suggesting that in deep layers semantic information is spread across many tokens. Furthermore, averaged representations give better predictability scores than concatenated ones, probably due to the ablation of semantically irrelevant positional information. We finally identify semantic layers in vision transformers by processing pairs of images that share the same class. These semantic layers are also the most informative about DeepSeek-V3 representations of descriptive image captions, where we observe significant information asymmetries between image and text representations.

## 1.1 Related work

**Representation alignment and convergence.** A growing body of work suggests that deep networks develop similar representational structure across architectures and modalities (Kornblith et al., 2019a; Sorscher et al., 2022; Maniparambil et al., 2024; Sucholutsky et al., 2024). This is formalized by the *Platonic representation hypothesis* (Huh et al., 2024), which postulates that, as model quality improves, representations converge to a shared structure. Evidence for this includes the finding that models can be stitched together (Lenc & Vedaldi, 2019; Bansal et al., 2021; Moschella et al., 2023) across architectures. However, these works largely treat representations as monolithic objects, without asking *where* in the network shared structure emerges, or *how much* information one representation carries about another.

**Multilingual representations.** The related question of whether multilingual LLMs develop language-independent representations was partially studied by searching for a linear transformation between simple concepts expressed in different languages (Peng & Søgaard, 2024) and by using Sparse Autoencoders (Brinkmann et al., 2025). Another complementary line of research studied the role of English acting as an internal pivot language using logit lens (Wendler et al., 2024) and neuron importance scores (Zhao et al., 2024), to try to explain the observed representation similarities across languages. Cheng et al. (2025) study LLM representations layer by layer using, among other tools, the Information Imbalance, focusing solely on last-token representations of the same text across models.

**This work.** By measuring the Information Imbalance across different models, layers and token aggregations, we observe that semantic information is spread across many tokens of deep internal representations, and we quantify the *directed* predictability between representations of translations, images sharing a class, and image-caption pairs, unifying and extending recently observed results.

## 2 Methods

### 2.1 Representation choice

**Text.** LLMs encode sentences (or other text units) as a sequence of tokens. What we refer to as the semantic content (or meaning) of a sentence should be a global property emerging from the interaction of all the tokens present in it, encoded in the respective representations. However, in deep layers, the attention mechanism can move information across tokens, and eventually concentrate it in a specific token, as indeed happens to some extent in the last token of autoregressive LLMs. Thus, it is natural to ask how the choice of token representation aggregation affects the observed representation alignments. In Sec. 3.1.1, we consider using the last token representation as a proxy for the whole sentence as in, for example Cheng et al. (2025) and Yin et al. (2024), and compare it to two different ways of aggregating token sequences: their average, a standard pooling technique used in, for example, Huh et al. (2024) and Valeriani et al. (2023), and their concatenation. Concatenating tokens is computationally expensive and thus is only feasible for moderate sentence lengths. Its main property is to preserve positional information, and it was used in Joshi et al. (2020) to predict spans of masked tokens using the concatenation of their left and right boundary tokens. Given that the averaged representation provided the best alignment scores between translations, we used it in all subsequent experiments.

### 2.2 Data and models

**Translations.** We use samples from Opus Books (Tiedemann, 2012), with pairs of sentences with lengths between 40 and 80 tokens, extracted from different novels. We excluded the last two tokens of each sentence from the analysis, as they consistently correspond to punctuation marks (e.g., a period or a period followed by a quotation mark), which introduce trivial similarities in the representations. We consider between 1000 and 3000 pairs of sentences where one of the languages is English, and the other is one of Spanish, Italian, German, French, Dutch, or Hungarian. This set was chosen based on their presence in the repository with more than a thousand samples after filtering, support by the language models we use, and our ability to manually check translation quality. For text representations, we focus on DeepSeek-V3, a MoE model with

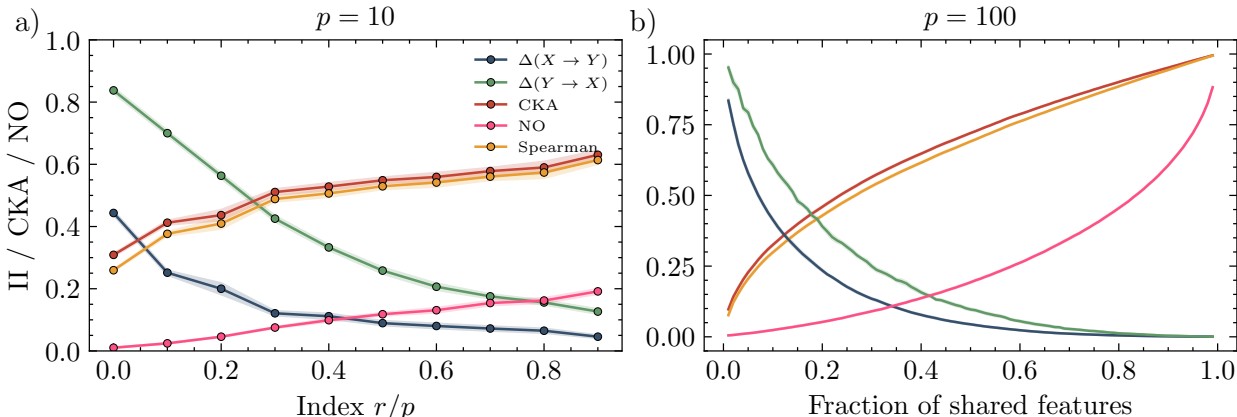

Figure 1:  **a)** Information Imbalance (II) $\Delta(X \to Y)$ and $\Delta(Y \to X)$, CKA, Representation Similarity Analysis (RSA) and Neighborhood Overlap (NO) for a synthetic Gaussian construction in which each index $r$ generates a pair $(X_r, Y_r)$ via $Y_r = B_r X_r + \varepsilon$, with $X_r \sim \mathcal{N}(0, I)$, $\varepsilon \sim \mathcal{N}(0, \sigma^2 I)$, in $p = 10$ dimensions. The matrices $B_r \in \mathbb{R}^{p \times p}$ have monotonically increasing rank, from $r{=}1$ to full rank at the final index. Note that *smaller* II means more predictivity, whereas *higher* CKA, RSA, and NO mean higher similarity. The Information Imbalance is the only measure able to reveal that $X$ is more informative than its noisy transform $Y$. **b)** Information imbalance and other metrics on a high-dimensional Gaussian vector with $p = 10^2$ components. We compute II, CKA, RSA, and NO with respect to a smaller vector with only a fraction of the $p$ features. In both panels, shaded bands (often barely visible) show the standard error over 10 jackknife repetitions. Sample size is 2500.

671B total parameters (DeepSeek-AI et al., 2025); to compare it with a widely used family of smaller models, we also use LLama3 (Meta, 2024) with 1, 3 and 8 billion parameters.

**Images.** We use the ImageNet-1k dataset (Deng et al., 2009), sampling 1,000 same-class pairs without replacement (to ensure we have no duplicates in the data), as a proxy for semantically related content. Thus, in this dataset, each pair is given by two different instances of the same ImageNet-1k category. We process the data with image-gpt-large (Chen et al., 2020) and DinoV2-large (Oquab et al., 2024). These models use fundamentally different training objectives. Image-gpt performs next-token prediction after down-scaling, row-wise unrolling, and color quantization of the input image, closely mimicking the autoregressive strategy of LLMs; previous work suggests that its semantic representations peak in the middle of the network (Chen et al., 2020). DinoV2 trains a student network to match a teacher on different augmentations of the same image, producing representations whose final layer is designed as the input to downstream tasks such as depth estimation and image segmentation.

**Image-caption pairs.** We use the Flickr30k dataset (Young et al., 2014), in which each image is paired with five human-generated captions that we concatenate into a single description. Images are processed with image-gpt-large and DinoV2-large; captions are encoded with DeepSeek-V3. We additionally include the ViT-L/14 visual encoder for images and the masked self-attention transformer for captions of CLIP (Radford et al., 2021) and three DinoV2 variants (small, base, large) to probe the effect of joint text-image contrastive training and vision model scale, respectively.

## 2.3 Quantifying the relative information content between representations

Our analysis is focused on the quantification of the relative information content of deep representations of translations of the same sentences, images of same-category objects, and image-caption pairs. We perform this analysis using the Information Imbalance (Glielmo et al., 2022; Del Tatto et al., 2024), a statistics which (qualitatively) measures the capability of a representation to predict another representation. Before discussing our main results, we present II measurements on synthetic data distributions. For a comparison, we

analyze the same data with other standard statistics used to compare different representations, in particular the Central Kernel Alignment (CKA) (Kornblith et al., 2019b) and the Neighborhood Overlap (Huh et al., 2024).

The Information Imbalance compares the neighborhood structures of two representation spaces $X$ and $Y$. Given representations $\{\mathbf{z}_i^X\}_{i=1}^N$ and $\{\mathbf{z}_i^Y\}_{i=1}^N$, we compute all pairwise distances and rank points $j \neq i$ in each space by increasing distance, obtaining ranks $r_{i,j}^X$ and $r_{i,j}^Y$. The Information Imbalance from $X$ to $Y$ is the average normalized average rank in $Y$ of the nearest neighbor of $i$ in $X$:

$$\Delta(X \to Y) = \frac{2}{N-1} \frac{1}{N} \sum_{i=1}^N r_{i,j^*(i)}^Y,$$

where $j^*(i)$ is the index of the nearest neighbor of $i$ in space $X$, i.e., the unique $j$ such that $r_{ij}^X = 1$. If neighborhoods in $X$ predict neighborhoods in $Y$, these ranks are small and $\Delta(X \to Y)$ is close to zero; if $X$ carries no information about $Y$, the expected rank is uniformly distributed and $\Delta(X \to Y) \approx 1$. The asymmetry $\Delta(X \to Y) \neq \Delta(Y \to X)$ quantifies directional predictability across models or modalities (Glielmo et al., 2022).

We start by providing two examples in synthetic Gaussian setups to give an intuition about the measurement of representation similarity using the II, CKA (Kornblith et al., 2019b), the Representation Similarity Alignment (RSA), i.e., Spearman's rank correlation (Spearman, 1904), and the Neighborhood Overlap (NO) measure from Huh et al. (2024), and show how information asymmetries can emerge. We first compute these metrics for a synthetic Gaussian construction in which each index $r$ generates a pair $(X_r, Y_r)$ via $Y_r = B_r X_r + \varepsilon$, with $X_r \sim \mathcal{N}(0, I)$ and $\varepsilon \sim \mathcal{N}(0, \sigma^2 I)$, in $p = 10$ dimensions. The $B_r$ matrices are defined by first choosing a target rank $r \in \{1, \dots, D\}$ and then sampling independent Gaussian factors $U_r \sim \mathcal{N}(0, 1)^{D \times r}$ and $V_r \sim \mathcal{N}(0, 1)^{r \times D}$, with the linear map given by $B_r = U_r V_r$, which enforces a transformation of rank $r$. High–rank settings preserve most of the structure in $X_r$, whereas low–rank settings collapse information into a low-dimensional, noisy subspace. In Fig. 1 a), we show that the Information Imbalance captures both the strength and the direction of the relative information across layers in this controlled synthetic setting. In particular, the Information-Imbalance shows that the predictivity between $X$ and $Y$ is *directional*. Indeed, $\Delta(X \to Y)$ is significantly lower than $\Delta(Y \to X)$. For comparison, we also report CKA and the Neighborhood Overlap, which track the overall loss of alignment between $X_r$ and $Y_r$ but, being symmetric, cannot reveal the asymmetry captured by Information Imbalance.

In Fig. 1 b), we compute the Information Imbalance between a Gaussian vector with $p = 10^2$ components and a smaller dimensional vector containing only a given fraction of them, to again highlight how the Information Imbalance captures the asymmetry, and to obtain a benchmark of its range of values in a reference scenario. In Appendix B, we compare the metrics for $p = 10^2$ and $p = 10^5$, confirming the statistical power of the Information Imbalance.

## 3 Results

### 3.1 Comparing translation representations

We now present results of a set of analyses performed using the Information Imbalance (II) on deep representations of data points that qualitatively convey approximately the same information to a human. We start by considering translations of the same sentence in different languages.

#### 3.1.1 Representation choice: Last token, concatenated tokens or averaged tokens?

Fig. 2 a) shows the II between English and Italian translations from the Opus Books parallel corpus collection, using three different representations: (i) The last token, (ii) the concatenation of the last $T$ tokens, or (iii) their average. Note that -under the Platonic representation hypothesis- for an ideal dataset of translations, an ideal network separately processing the sentence pairs should generate equivalent representations, namely, somewhere in the network, the II from one language to the other should be close to zero. Thus, we expect better abstract representations to have a lower Information Imbalance. In all three cases, the II is

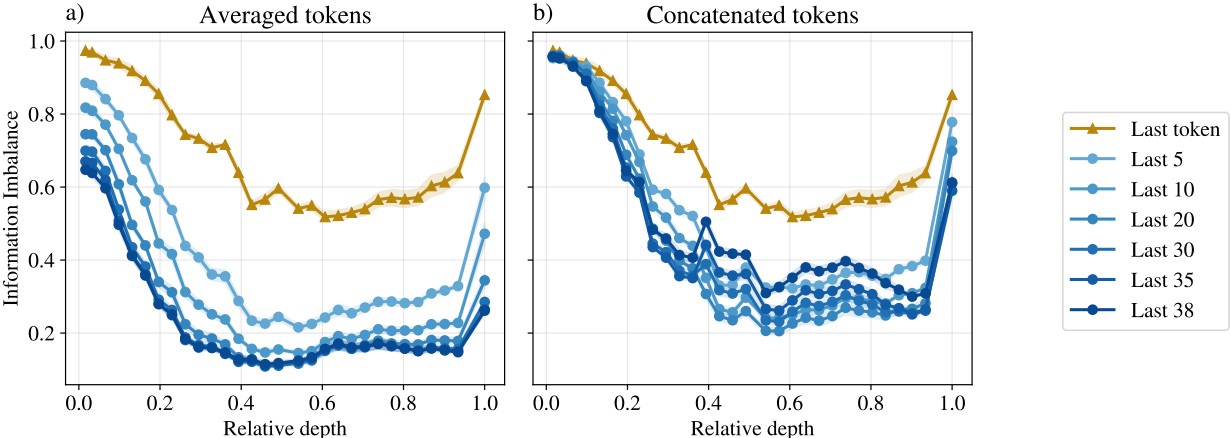

Figure 2: **Representation choice.** Information Imbalance from English to Italian tranlsations in DeepSeek-V3 when tokens are a) averaged or b) concatenated. The (hardly visible) shaded colored areas correspond to the standard deviation obtained by subsampling half of the samples five times.

minimum in central layers, that were independently found to be related to an abstraction phase(Cheng et al., 2025), simultaneously far away from the encoding ($l = 0$) and decoding ($l = L$) layers, which necessarily contain more language-specific information. We highlight that the predictability between representation significantly increases from single-token to many-token representations. Given that the observed neighborhood predictability seems driven by semantic correspondences, this result suggests that semantic information is spread across many tokens, and not concentrated in, say, only the last token.

Among the two considered aggregation methods, i.e., concatenation and average of the last $T$ tokens, we note three main differences. In initial layers, the average of an increasing number of tokens increases predictability (reducing II) between representations with respect to the last token alone, whereas this does not happen for the concatenated representation. This is consistent with the fact that the meaning of sentences, which drives predictability of the representations, does not rely on precise positional information and it will be typically distributed across different tokens in a sentence and its translation. Furthermore, we observe that, in deep layers, the average of a larger number of tokens in the representation monotonically improves predictability (whereas this does not happen for the concatenated representations) and, most importantly, gives the "best" (i.e., lowest) predictability score. Consequently, in what follows we focus on token-averaged representations.

For all curves, predictability is still larger at the last layer than at the first. This is plausibly due to the fact that the representation of the last layer must still contain some of the semantic information computed in the central layers, which must be at least partially used for next-token prediction. As a null hypothesis, we show in Appendix C that performing batch-shuffling on any of the datasets leads to completely uninformative representations (Information Imbalance gives 1 at every layer), since the semantic correspondence between translations is destroyed. Appendix E shows that the results obtained in Fig. 2 qualitatively agree with those given by CKA and Neighborhood Overlap, highlighting some limitations of these metrics.

Note that, here and below, we are focusing, for simplicity and visualization purposes, on the II of equal-depth representations. In Appendix F, we present II results across different depths. In general, we observe that late layers tend to be more predictive about earlier layers than the reverse (thus providing another systematic source of information asymmetry). The central layers, which we are conjecturing to be the main locus of semantic processing, are also those displaying the maximal cross-layer predictability scores.

### 3.1.2 Information Imbalance between different languages and different models

Fig. 3 a) shows that the II profiles across layers for Spanish, Dutch, German, and French translations are in very good agreement with the Italian one. Hungarian, which is simultaneously the tested language with

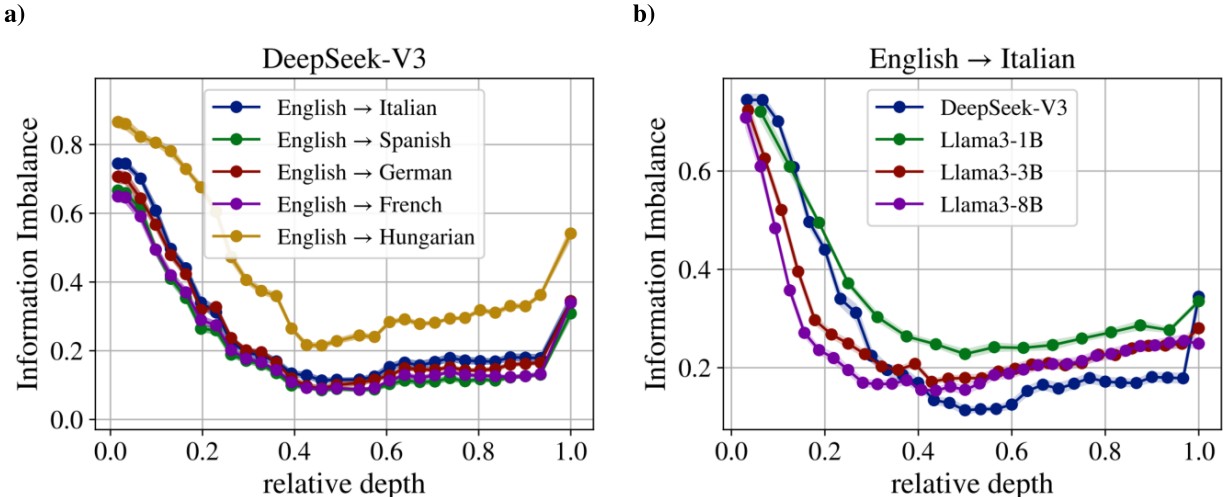

Figure 3: **Comparison with other languages and models.** Panel a): Information Imbalance from English to several languages, computed on representations generated by DeepSeek-V3. Panel b): Information Imbalance from English to Italian, computed on representations generated by Llama3 models with 1,3, and 8 billion parameters, and by DeepSeek-V3 for comparison. In both panels we used the average of the last 20 tokens to represent sentences, and the (hardly visible) shaded areas correspond to the standard deviation obtained by subsampling half of the samples five times.

the smallest amount of resources available on the internet[1] and the only non-Indo-European language in our set, shares the same qualitative II profile, but with larger II values (smaller predictability). In Fig. 15 of Appendix G, we take a closer look at this qualitative observation: the figure shows a significant correlation between the amount of online content available for a language, used as a rough proxy of its presence in training corpora, and the minimum Information Imbalance between English and that language, for an extended set of languages. We stress, however, that this result does not establish a causal link between language exposure and predictability scores, but a first piece of correlational evidence between these two quantities.

Fig. 3 b) shows instead that the II layer profile for English-Italian translations is similar for the Llama3 family of models with 1, 3 and 8 billion parameters. DeepSeek-V3 results from Fig. 2 a) are also shown for direct comparison. Notably, and in accordance with the Platonic representation hypothesis, increasing model size reduces the II, predominantly in the inner layers. The position of the global II minimum also shifts slightly towards earlier layers for the larger models. We highlight that, qualitatively, the II profiles of all models are very similar, and DeepSeek-V3's global minimum is deeper than for the other models. Fig. 14 in Appendix G shows that the II layer profiles are also similar for Qwen3-8b, for an extended set of languages.

### 3.1.3 Information asymmetries

Given that English is the dominant language in terms of training resources and performance (Zhu et al., 2024), one can ask if English representations are more informative than those generated processing other languages. Fig. 4 a) shows that this asymmetry is in fact observed, most notably in the last quarter of the network, for the specific case of English-Italian translations. Nonetheless, the II asymmetry profile is far from trivial. Notably, both representations are equally predictive of each other where predictability is maximal, around the middle of the network. This is in line with the hypothesis that the middle layers store semantic information that might be largely language-independent. Moving towards earlier layers, at around 0.2 relative depth, we see that a small gap opens between the two II curves, where again English is the most informative, whereas in the very first layers we observe a small inverted asymmetry. Fig. 4 b) shows the II

---

[1]https://en.wikipedia.org/wiki/Languages_used_on_the_Internet

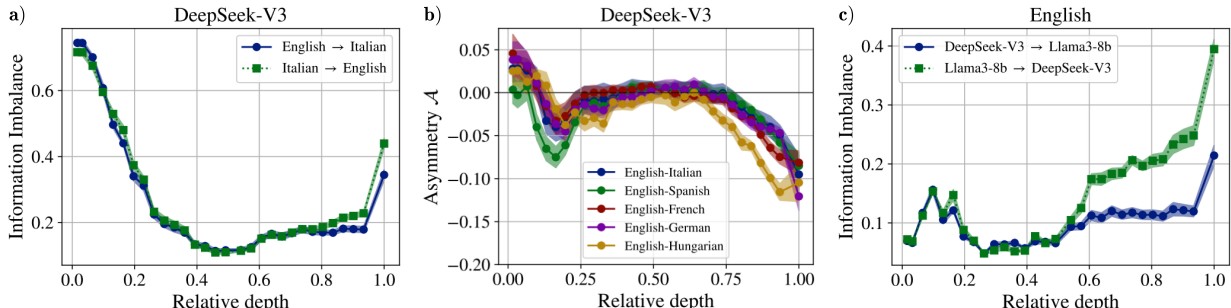

Figure 4: **Information asymmetries.** Panel a): Information Imbalance (II) from English to Italian and from Italian to English, computed on representations generated by DeepSeek-V3. Panel b): II Asymmetry $\mathcal{A} = II(English \rightarrow other) - II(other \rightarrow English)$ between English and other languages, computed on representations generated by DeepSeek-V3. Note that, under this definition of asymmetry, a negative value implies that English is more informative than the other language. Panel c): Information Imbalance between representations generated by DeepSeek-V3 and Llama3-8b on the same English text, as a function of comparable relative depth in both models. The shaded area corresponds to one standard deviation, computed with a Jackknife procedure by subsampling five times half of the samples. In all three panels, we used the average of the last 20 tokens.

asymmetry $\mathcal{A} = II(English \rightarrow other) - II(other \rightarrow English)$ across the layers of DeepSeek-V3. Note that, under this definition, a negative asymmetry corresponds to the case in which English representations are more informative than representations in the other language. We observe that the asymmetry profile across layers is robust across translations in five different languages, with the same overall trend: the representations of English sentences are asymmetrically more informative in the early and late layers. The very first layers show a consistent positive asymmetry, where English tokens are less predictive than those in the other language, which could be an effect of the tokenization, and is left for a better understanding in future work.

Fig. 9 in Appendix D shows that the corresponding asymmetry profiles across the layers of Llama3-8b also are positive in the very early layers, approaching zero in middle layers, and negative in the last quarter of the network, with the exception of the last layer, in which the asymmetry is roughly zero.

One can also wonder if the representations of a huge model such as DeepSeek-V3 are more informative than those generated by a smaller Llama3 model. For identical English inputs, Fig. 4 c) shows the II between DeepSeek-V3 and Llama3-8b representations at comparable relative depths. In contrast to the II profile between translations in Fig. 2, in Fig. 4 c) the II in the very first layers is very small, suggesting that the word embeddings of the two models are very similar. The II has then a small peak at roughly 0.1 relative depth, plausibly related to some difference in input processing between the two networks. Afterwards, at intermediate depths, we observe a region of maximal mutual predictability (lowest II) without information asymmetries up until roughly half of the networks. Again, this is plausibly related to the fact that the middle layers are those that contain the most "universal" semantic information. In the second part of the network, we observe a massive gap between the curves, where DeepSeek-V3 representations are far more predictive than those from Llama3-8b.

### 3.1.4 Token-token Information Imbalance

We have used representations constructed out of the average of several tokens to measure mutual predictability between translations (Figures 2 and 3) and to quantify information asymmetries between English and other languages and between different LLMs (Fig. 4). In this section, we zoom-in on two languages and look at the correlation structure between tokens, trying to gain further qualitative insights.

Fig. 5 shows the II between the last token and a previous token at distance $\tau$ from it, for English and Italian representations generated by both DeepSeek-V3 and Llama3-8b. For DeepSeek-V3, in the very first layers,

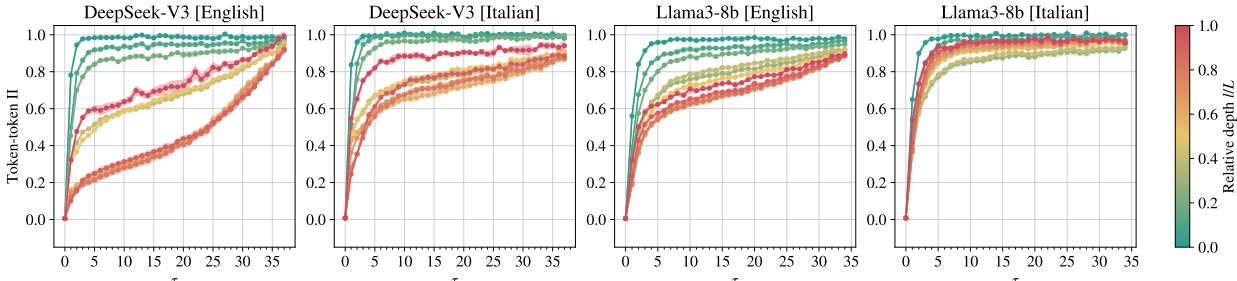

Figure 5: **Token-token Information Imbalance.** Information Imbalance from the last token to a previous token at distance $\tau$, as a function of $\tau$. Panels a) and b) correspond to DeepSeek-V3 representations of English and Italian text, whereas panels c) and d) correspond to Llama3-8b representations of English and Italian. The (hardly visible) shaded area corresponds to one standard deviation, computed with a Jackknife procedure by subsampling five times half of the samples.

the last token is, on average, already not informative about a previous token at distance $\tau \approx 5$. In deeper layers, tokens become more and more informative about each other. The layers with strongest correlations (lowest II) are those with relative depth roughly between 0.6 and 0.9, which approximately coincides with the broad region with very low II between translations in Fig. 2, right after the global minimum. The token-token predictability in these deep layers is much larger in DeepSeek-V3 (Figs. 5 a) and 5 b)) than in Llama3-8b (Figs. 5 c) and 5 d)). This analysis is to be compared with the one of Fig. 3 b), where DeepSeek-V3's II between translations is lower (higher predictability) than the II computed on any of the Llama3 models. Furthermore, by comparing Fig. 5 a) and Fig. 5 b), we see that the token-token II in very deep layers of DeepSeek-V3 is also much lower in English than in Italian, and the same happens for Llama3-8b between panels c) and d). Thus, we qualitatively observe that representations that are more predictive across languages are also those in which the information is more consistent across many tokens, similarly to the observed information asymmetries from Fig. 4, and to Hosseini & Fedorenko (2023), who showed that better performing models tend to display linear correlations across longer spans of tokens, giving straighter token trajectories.

## 3.2 Comparing image and image-text representations

### 3.2.1 Convergence in representations of same-class images

We first compare representations for pairs of images that share the same ImageNet-1k class, but depict different instances. For example, one pair might consist of images of two *black widows*, another of two *accordion* pictures. The representations of these image pairs should be close in layers that capture semantic information. We use mean-token activations to represent each image, consistently with previous work by Valeriani et al. (2023). Since the assignment of the two images to the two representation spaces is arbitrary, the Information Imbalance is symmetric, except for statistical errors. Therefore, we report its average between the two directions. Results are shown in Fig. 6 a). For DinoV2, the Information Imbalance decreases monotonically and reaches values close to 0.35 around its last layer. For image-gpt-large, the minimum ($\approx 0.65$) is attained around 42% relative depth, after which the Information Imbalance rises again, reaching $\approx 0.85$ at the last layer. This is consistent with the respective training objectives. DinoV2 is trained so that its final-layer representations serve as input to downstream tasks such as depth estimation and segmentation, which should lead to rich semantic content being encoded at the output. Image-gpt-large, an autoregressive model, concentrates instead semantic content in the middle of the network, and probably reverts to lower-level, pixel-predictive features near the output. The contrasting profiles of DinoV2 and image-gpt-large suggest that the training objective is shaping where in the network semantic information is most concentrated, with autoregressive models (such as the textual transformers and image-gpt-large) placing it in intermediate layers, and encoder models pushing it toward the final layers.

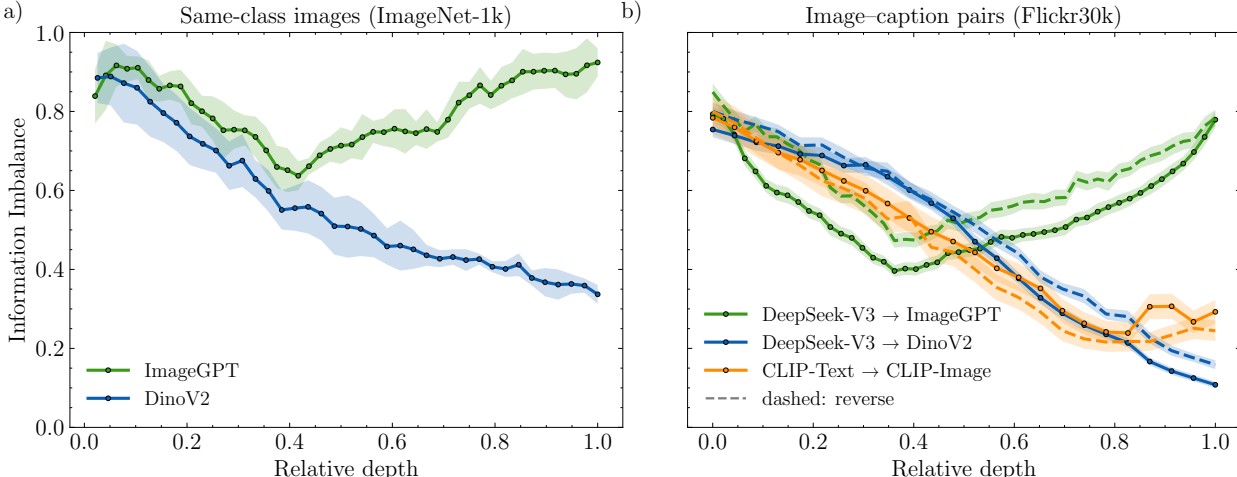

Figure 6:  **a)** Within-model Information Imbalance for 1,000 same-class image pairs from ImageNet-1k, using mean-token activations. For each model, we compare the representations of two distinct images of the same class at the same layer, and plot the result as a function of relative depth. DinoV2 reaches its minimum at the last layer; ImageGPT at $\approx 42\%$ relative depth. **b)** Cross-modal Information Imbalance on Flickr30k image–caption pairs. One model is held at a fixed layer (DeepSeek-V3 at $\approx 60\%$ relative depth, or CLIP-Image at its last layer) while all layers of the partner model are swept. Solid lines show the Information Imbalance in the $A \rightarrow B$ direction; dashed lines show the reverse direction ($B \rightarrow A$), keeping the same model fixed at the same layer.

### 3.2.2 Images and captions: multimodal data sharing semantic content

Image-caption pairs convey approximately the same meaning in different modalities, similarly to translations of the same sentence in different languages. The within-model image–image analysis of Fig. 6 a) identifies, for each vision model, the layers where semantic information is most concentrated. We expect these same layers to yield the strongest cross-modal alignment with textual representations of the corresponding captions. In Fig. 6 b), we test this by fixing DeepSeek-V3 at $\approx 60\%$ relative depth (in the region that minimizes the Information Imbalance in the translation analysis of Fig. 2) and sweeping all layers of each vision model on the Flickr30k dataset. We compute the Information Imbalance in both directions: the gap between the two reveals how asymmetric the cross-modal relationship is, serving as a measure of how much richer one representation is relative to the other. To verify that the results are not sensitive to this particular choice, Fig. 16 c) of Appendix H shows the Information Imbalance from representative layers of the vision models against all DeepSeek-V3 layers, showcasing how the values stabilize after 60% relative depth.

For DinoV2, the minimum Information Imbalance (DeepSeek-V3 $\rightarrow$ DinoV2) is $\approx 0.10$, attained at the final layer; in the reverse direction (DinoV2 $\rightarrow$ DeepSeek-V3), the minimum is $\approx 0.30$. For image-gpt-large, the minimum Information Imbalance from DeepSeek-V3 is $\approx 0.40$ (at $\approx 40\%$ relative depth), while in the reverse direction it reaches $\approx 0.60$. The substantially higher values compared to DinoV2 indicate weaker cross-modal alignment. Note that both the curve profiles and the absolute Information Imbalance values are largely compatible with the image–image results (Fig. 6 a). This suggests that the nature and quality of processing by the visual transformers is the dominating factor in determining image–text convergence. Note, moreover, an asymmetry in favor of DeepSeek-V3 being more predictive of the visual models than the opposite (dashed lines above solid lines in the plot). We also include in Fig. 6 b) the ViT-L/14 visual encoder ($\approx$303M parameters) of the multimodal CLIP system (Radford et al., 2021), paired with its autoregressive text encoder ($\approx$123M parameters). Note that both CLIP encoders are orders of magnitude smaller than DeepSeek-V3 (671B parameters), while DinoV2-large ($\approx$300M parameters) is comparable in size to the CLIP visual encoder.

The CLIP pair reaches a minimum Information Imbalance of $\approx 0.25$, noticeably higher than the DeepSeek-V3 $\rightarrow$ DinoV2 value of $\approx 0.10$. Two independently trained models (DeepSeek-V3 and DinoV2) thus achieve stronger cross-modal alignment than a pair explicitly trained to align image and text representations. This is consistent with Norelli et al. (2023), who show that frozen unimodal encoders can be aligned *post hoc* through a small set of paired "anchors" in order to match CLIP-level zero-shot performance. This should however not be read as evidence that scale always dominates joint training: image-gpt-large, the largest vision encoder we consider (1.4B parameters), yields the weakest alignment of the three pairings ($\approx 0.40$ when paired with DeepSeek-V3). The ordering instead reflects the within-modality semantic alignment of each vision encoder as reported in Fig. 6a (DinoV2 reaches $\approx 0.35$, image-gpt-large plateaus near $\approx 0.65$), suggesting that size, training objective and architecture all play a role in determining the semantic content of the representation that each encoder develops. To rule out the possibility that the gap between DeepSeek-V3 $\rightarrow$ DinoV2 and CLIP is driven by CLIP's limited token context width, we verified on the subset of images whose 5-caption concatenation fits within CLIP's 77-token context that CLIP-text$\leftrightarrow$CLIP-image still reaches an II of $\approx 0.18$, well above that of DeepSeek-V3 $\rightarrow$ DinoV2's $\approx 0.11$ on the same subset.

Two further differences stand out. First, multimodal training produces more uniform alignment across layers: the CLIP curves remain flatter than those of the non-aligned pairs, indicating that contrastive training encourages cross-modal predictability throughout the network rather than concentrating it in a narrow depth range. Second, the CLIP pair exhibits a more symmetric Information Imbalance between the two directions than the DeepSeek-V3/vision transformer pairs, which may reflect the explicit CLIP alignment objective. That two very different cross-modal comparisons, namely (i) a large independently trained LLM–ViT pair and (ii) a much smaller jointly trained pair, both reach minimum Information Imbalance values in the range 0.2–0.3 raises the intriguing question of whether this threshold approaches an intrinsic ceiling in text-image alignment. We leave the exploration of this question to future work.

Finally, to probe the dependence of the observed cross-modal alignment on the scale of the vision model, we repeat the DeepSeek-V3–DinoV2 analysis with three DinoV2 variants (large, base, and small); see Fig. 16 b) of Appendix H. The minimum Information Imbalance is 0.10 for DinoV2-large, 0.10 for DinoV2-base, and 0.15 for DinoV2-small. All three profiles reach their minimum at the last layer, confirming that this semantic region is robustly identified regardless of vision model size. Moreover, similarly to what we observe with the results on translations of Fig. 3, we see that bigger model sizes produce lower values of Information Imbalance. For these vision models, the small gap between the large and base variants suggests that the quality of cross-modal alignment may saturate once the vision model reaches a sufficient capacity. We repeat the analysis with a smaller LLama3.1-8B model, confirming the same qualitative results we obtain for the larger DeepSeek-V3 model. We report the results in Appendix H.

## 4    Conclusion

The recent observations on representation alignment by Huh et al. (2024) call for thorough analyses determining how different metrics affect their results (Gröger et al., 2026), and more generally, how presumably universal semantic information is encoded in deep neural networks. We contributed to this research line by measuring the relative information content between representations generated by deep neural networks processing inputs in different languages and modalities that carry approximately the same meaning. For these purposes, we used the Information Imbalance (II), an *asymmetric* measure based on neighborhood ranks. First, using synthetic datasets, we showed that the II captures information that standard similarity metrics, such as Centered Kernel Alignment, Representation Similarity Analysis, and Neighborhood Overlap, cannot reveal. In contrast to these symmetric metrics, II provides a richer description of representation alignment by capturing a hierarchy in relative information content, while still being computed efficiently using distances between samples.

Then, using translated sentences, we explicitly characterized the layer dependence of the II inside a LLM with state-of-the-art performance, DeepSeek-V3. Consistently with recent experiments on cross-language retrieval (Liu & Niehues, 2025), and with semantic probing experiments (Cheng et al., 2025; Skean et al., 2025), we found that central layers representations for each language are most predictive about each other. Using the II, we were able to estimate the asymmetric information content of representations. We found

that the only layers in which mutual predictability across languages is symmetric are the central ones, while asymmetries arise in early and late layers. We explored how representation choice affects the observed predictabilities by comparing the II between (i) last-token representations, (ii) average, and (iii) concatenation of the last $T$ tokens. We found that averaged representations give the best scores (lowest II), probably because they partially remove positional information that is irrelevant for semantics, and that semantic information is spread across many tokens, instead of concentrated only in, say, the last token.

Regarding information asymmetries, we showed in Fig. 4 b) that English sentence representations are more informative than those generated by processing Italian, French, Spanish, German and Hungarian sentences, specifically in the second half of the network. Furthermore, we presented one case where DeepSeek-V3 representations are more informative than those generated by the smaller Llama3-8b in Fig. 4 c). At a qualitative level, we observed that such asymmetries also arise in token-token correlation structure, suggesting that long-range correlations could be a hallmark of representation quality, as also suggested by Hosseini & Fedorenko (2023). In Sec. G we investigated to what extent language exposure, roughly approximated by the online presence of different languages, can predict the observed alignment between translations, using an extended pool of languages.

In the visual domain, we observed that the autoregressive image-gpt-large model, when processing semantically-related images, produces a qualitatively similar II profile as autoregressive LLMs processing translations, where central layers are the most predictive. Instead, an encoder model such as Dino-V2 develops representations that are maximally predictive towards the last layers. We also observed that the layers with the highest relative information content for semantically-related images are also those where we observe the largest cross-modal convergence with DeepSeek-V3 textual representations.

Taken together, our results support the hypothesis of semantic convergence across languages, modalities and models, and establish a more nuanced picture of it. In particular, they suggest that convergence is a property of specific intermediate processing stages, that can differ from model to model (e.g., central vs. late layers), and even when different representations of semantically related inputs do converge, there might be significant asymmetries in their relative information content, which might depend on factors such as model size, training resources and objective, and modality.

In future work, we would like to more clearly disentangle the factors leading to asymmetric semantic content. Asymmetries in the information content of different distances defined on the same data space are potentially useful to perform feature selection. In a limiting case, if one distance measure is capable of predicting another distance measure, while the reverse is not true, the features characterizing the second distance measure can be discarded (Glielmo et al., 2022). This criterion has been used to reveal the arrow of time in high-dimensional time series (Del Tatto et al., 2024; Glielmo et al., 2022). We plan to extend these ideas to the analysis of latent representations of language, and identify and interpret the features which are responsible for symmetry breaking. We plan to investigate, moreover, if the lower convergence observed when vision models are involved is due to model size, training regime, or intrinsic properties of linguistic vs. visual inputs. More generally, we intend to study whether the asymmetry captured by the Information Imbalance behaves in the same manner as the asymmetry that could be captured by alternative methods such as linear mappings between representations, or by model stitching.

However, we believe that the ultimate question to be answered pertains to the *nature* of the semantic features that are shared across representational systems. For example, Tamkin et al. (2020) showed that linguistic information at different scales (word-level, sentence-level, document-level) is better decoded by classification probes using different frequency modes along the token axis. The fact that we achieved the best scores for alignment between translations by using mean token representations, which correspond to the zero-frequency mode, suggests that semantic information might be predominantly encoded in low frequency frequency modes. We hope that our contribution, by highlighting specific patterns of semantic convergence, lays the foundation for this kind of future explorations.

## 5 Acknowledgments

We thank Gemma Boleda and Corentin Kervadec for useful feedback. MM and MB received funding from the European Research Council (ERC) under the European Union's Horizon 2020 research and innovation program (grant agreement No. 101019291) and from the Catalan government (AGAUR grant SGR 2021 00470). AL, AM and SA acknowledge financial support by the region Friuli Venezia Giulia (project F53C22001770002). This paper reflects the authors' view only, and the funding agencies are not responsible for any use that may be made of the information it contains.

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

## A    The Information Imbalance as an upper bound on mutual information

Information Imbalance can be connected to information theory, as we briefly outline here, to provide a principled foundation for its use as a measure of representational predictability. For a full derivation of the formulae presented below, see the supplementary material of Del Tatto et al. (2024).

As shown in Glielmo et al. (2022), distance ranks are the empirical counterpart of copula variables (Nelsen, 2006), the continuous analogue of ranks in a probabilistic setting. Concretely, the copula variable associated to the distance rank $r_A^i$ of data point $j$ relative to sample $i$ in space $A$ is $c_A^i = \frac{1}{N} r_A^i$, and analogously for $c_B^i$.

Glielmo et al. (2022) and Del Tatto et al. (2024) defined the restricted mutual information $I^\varepsilon(R_A^i \to R_B^i)$ as

$$I^\varepsilon\left(R_A^i \to R_B^i\right) \;=\; -\int_0^\varepsilon d\tilde{c}_A^i \; H\left(c_B^i \,\middle|\, c_A^i = \tilde{c}_A^i\right), \tag{1}$$

where $c_A^i = \frac{1}{N} r_A^i$ are the copula variables computed from the distance ranks $r_A^i$ of data points, relative to a given data sample $i$ in space $A$ (and analogously for $c_B^i$), and $H(c_B^i \mid c_A^i = \tilde{c}_A^i)$ is the conditional entropy of

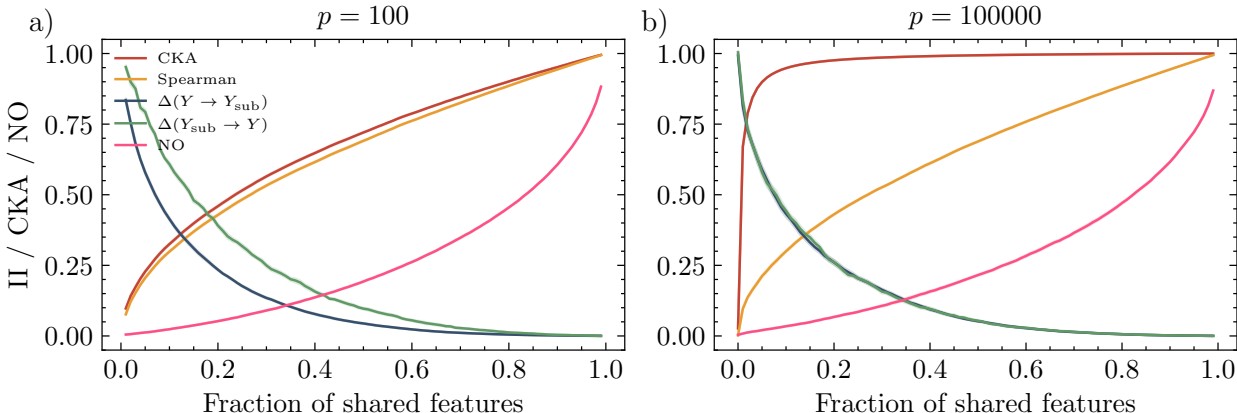

Figure 7: Information Imbalance, CKA, RSA, and NO computed between a $p$-dimensional Gaussian vector and a smaller vector retaining only a fraction of its components, for **a)** $p = 10^2$ (also shown in Fig. 1 b) and **b)** $p = 10^5$. As the dimensionality grows, CKA saturates near 1 once even a small fraction of components is shared, while the neighborhood- and rank-based measures (Information Imbalance, RSA, NO) retain discriminative resolution up to a much larger fraction of shared features. (Barely visible) shaded bands show the standard error over 10 jackknife repetitions. Sample size is 2500.

the $B$-space copula variable. In the limit $\varepsilon \to 0$, only the $k$ nearest neighbors in space $A$ contribute, and the Information Imbalance $\Delta(A \to B)$ (Eq. (1)), can be interpreted as an upper bound:

$$\Delta(A \to B) \;\gtrsim\; 2 \left\langle \exp\left( -\lim_{\varepsilon \to 0} \frac{I^\varepsilon\left(R_A^i \to R_B^i\right)}{\varepsilon} - 1 \right) \right\rangle_{i=1,\ldots,N} . \tag{2}$$

This bound becomes tight when the two representations are most aligned, confirming that the II is a principled, information-theoretically grounded measure of representational predictability. The tightness of this bound has been empirically validated in Gaussian settings by Umar et al. (2026).

## B   Comparing the Information Imbalance with other metrics

We compute the Information Imbalance and other similarity metrics between a Gaussian vector with $p$ components and a smaller-dimensional vector containing only a given fraction of them, for the cases of $p = 10^2$ and $p = 10^5$, in Fig. 7 a) and Fig. 7 b), respectively. For a $10^5$-dimensional Gaussian, CKA saturates close to 1 even when only a small fraction of the components is observed, while the Information Imbalance, the RSA, and the Neighborhood Overlap retain discriminative resolution up to a much larger fraction of shared features. This illustrative example echoes the findings of Huh et al. (2024) and Gröger et al. (2026), that observed that neighborhood-based metrics are preferable to CKA in high dimensions. We also highlight that, in Figs. 7 a) and 7 b), the Information Imbalance is more sensitive when there is a small fraction of shared features (weak signal) than the Neighborhood Overlap, and, in a complementary way, the opposite happens for a large fraction of shared features. In panel b), the information asymmetry between representations is captured by the gap between Information Imbalance curves, which becomes small when the dimensionality is large.

## C   Misalignment of translations erases semantic similarity

As a consistency check, Fig 8 shows the Information Imbalance for DeepSeek-V3 and Llama3.1-8b representations using misaligned translations, namely performing a batch-shuffle in one of the datasets. Since the semantic correspondence between sentences is destroyed, the representations are not informative about each

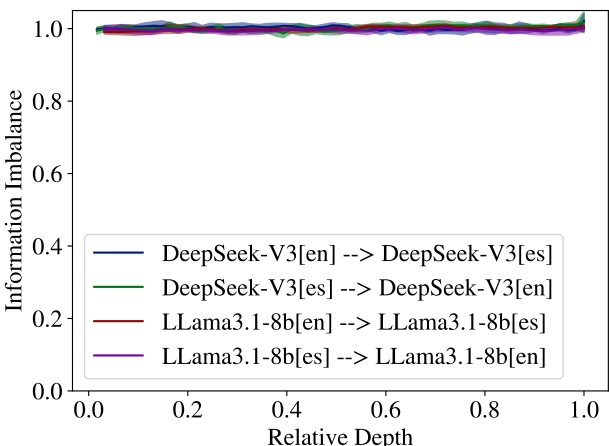

Figure 8: Information Imbalance between English (en) and Spanish (es) representations generated by DeepSeek-V3 and Llama3.1-8b, for the non-informative case, namely a misaligned dataset in which we batch-shuffle the Spanish translations. The (hardly visible) shaded area corresponds to one standard deviation, computed with a Jackknife procedure by subsampling five times half of the samples.

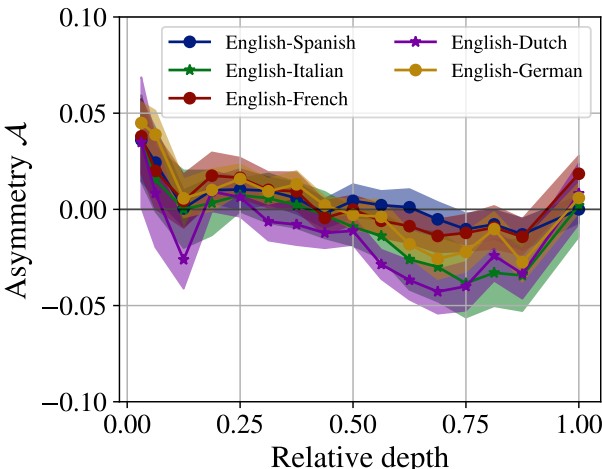

Figure 9: II Asymmetry $\mathcal{A} = II(English \rightarrow other) - II(other \rightarrow English)$ between English and other languages, computed on representations generated by Llama3-8b. Note that, under this definition of asymmetry, a negative value implies that English is more informative than the other language.

other, and thus the Information Imbalance is close to one, for all layers. The same occurs with misaligned image pairs and image-caption pairs (tested, but not reported here).

## D  Information asymmetry in translations processed by Llama3

Fig. 9 shows the Information Imbalance asymmetry $\mathcal{A}$ between English and other languages across layers. We see some similarities and some differences with respect to DeepSeek-V3's results from Fig. 4 b). As for DeepSeek-V3, Llama3-8b's II asymmetry is positive for the very first layers, roughly zero in middle layers, and negative around the third quarter of its relative depth. Contrary to Fig. 4, the last layer of Llama3-8b presents roughly zero asymmetry, and there is no negative local minimum of $\mathcal{A}$ in between the very first layers and the central layers.

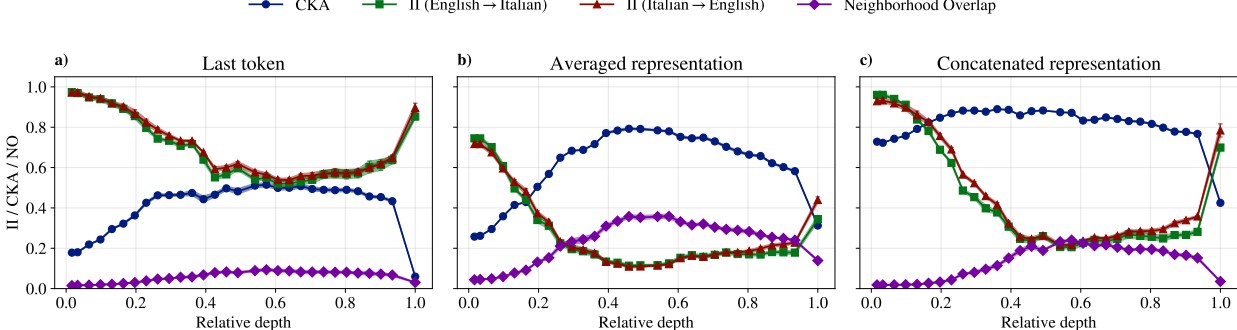

Figure 10: **Central Kernel Alignment (CKA) and Neighborhood Overlap (NO) comparison on translations.** Information Imbalance (II) from English to Italian and from Italian to English, together with CKA and Neighborhood Overlap computed on the same activations, using a) the last token, b) the average of the last 20 tokens, and c) the concatenation of the last 20 tokens. Note that for II, *lower* values cue higher alignment, whereas for CKA and NO *higher* values cue higher alighnment.

## E  CKA and Neighborhood Overlap comparison on translations

Fig. 10 compares the II curves between English and Italian translations as a function of DeepSeek-V3's relative depth with CKA and Neighborhood Overlap (NO) computed on the same data, using a) the last token, b) the average of the last 20 tokens, and c) their concatenation.

First, we highlight that, for all three representational choices, the three metrics concurrently achieve their maximum score (lowest II, maximum CKA and NO, respectively). This confirms the robustness of the result we report in the main text about the presence of a central phase where representations are most clearly encoding shared semantic information.

However, there are several nuances to note. Starting with the Neighborhood Overlap, we note that it only reaches maximum values of roughly 0.1, 0.4 and 0.2 in panels a), b) and c), respectively, far from its theoretical maximum of 1, suggesting it is underestimating representational convergence. The last-layer Neighborhood Overlap is barely above zero for last-token and concatenated representations, with values of $0.031 \pm 0.002$ and $0.035 \pm 0.003$, respectively. Small alignment values were also observed and even highlighted as a limitation of the convergence of representations in Huh et al. (2024), although the authors do not provide arguments to understand the reasons behind this effect. We note that, if the $k$-th neighbor of representation $A$ is instead the $k+1$ neighbor of representation $B$, automatically that point counts as *outside* the $k$-neighborhood, even if it is extremely close, rendering NO alignment values very low. This does not affect Information Imbalance, which, instead of evaluating whether $k$ neighbors are shared, measures what is the *rank* in space $B$ of $k$ neighbors in space $A$, and vice-versa.

CKA displays much larger alignment values than NO in all three panels, with the exception of the last layer of panel a), where $CKA = 0.059 \pm 0.001$, as the measure struggles to capture the weak signal provided by last-token representations. Moreover, while the CKA similarity profiles are compatible with those computed with II and NO in panels a) and b), we observe that CKA estimates very high similarity in the initial layers of panel c), in complete contrast with the other two metrics. We hypothesize this to be a spurious artifact of working in extremely high dimensions (20 concatenated tokens, each embedded in the 7,168 dimensions of DeepSeek-V3), that is avoided by the metrics based on neighborhood ranks.

## F  Predictability between different layers

For simplicity, in Sec. 3.1 we presented our results constrained to equal-depth comparisons. Here, we show examples of how the II between different layers behaves. For clarity, we distinguish three scenarios, namely:

comparing different layers from the same model processing the same sentences in English (Sec. F.1), comparing different layers from different models processing the same sentences in English (Sec. F.2), and comparing different layers from different models, each processing a different language (Sec. F.3).

We highlight that the following results are limited to two models. Furthermore, asymmetry could also be captured by model stitching and linear mappings, and it would be interesting to see if all these methods display the same asymmetries. We plan to address these issues in future work.

### F.1 Same language, same model, different layers

Fig. 11 shows the II between a fixed layer of a fixed model processing English sentences and all other layers. As fixed layers of interest, we took the first, the central and the last layers of DeepSeek-V3 and Llama3-8b, shown in panels a) and b), respectively (the patterns are qualitatively similar for the two models). We note that, in this setup, the II from any layer to itself vanishes, since we are not changing language nor model. In line with our results in the main text, the central layers are generally good predictors of the other layers, largely symmetrically so. Interestingly, a clear asymmetry emerges between the late and early layers, with late layers much better at predicting the early ones than the reverse. We conjecture that this stems from two factors. On the one hand, the last layer, like the early ones, must contain concrete token-related information in order to guess the next word. On the other, it must also encode semantic information that was extracted in the central layers, in order to refine its guess. These combined factors could make the late layer an asymmetrically good across-the-board predictor. Future work should evaluate this conjecture more thoroughly.

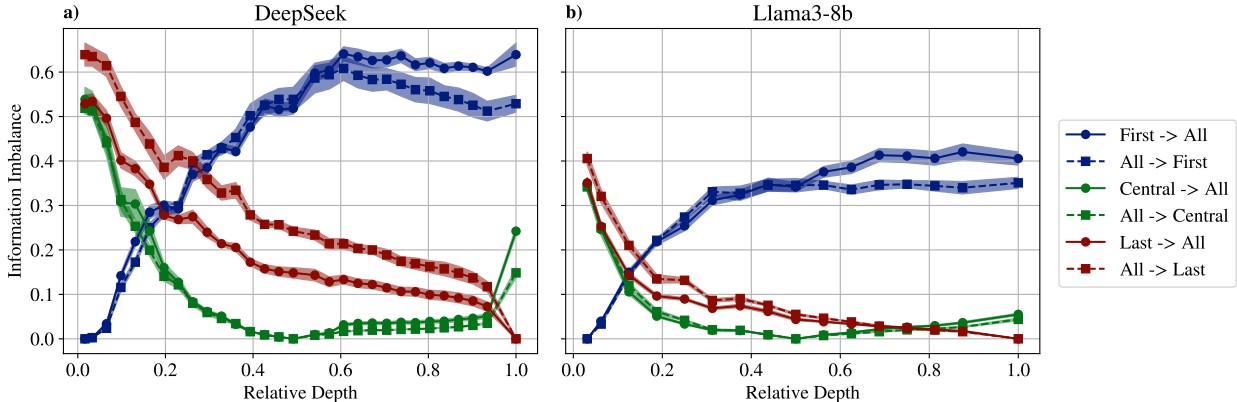

Figure 11: **Predictability between different layers, fixing model and language.** Information Imbalance between representations from a fixed-layer and all other layers, as a function of relative depth, for DeepSeek-V3 (panel a)) and Llama3-8b (panel b)) processing the same English sentences. As representative layers of different processing stages, we picked the *first*, *central* and *last* layers of each model.

### F.2 Same language, different model, different layers

Fig. 12 shows the II between a fixed layer of one model and all layers of other model, both processing the same sentences in English. Again, the central layers are generally good predictors across the board. In contrast to the fixed-model case, here there are two contributions to information asymmetry, namely: the already seen late- vs. early-layer effect and stronger vs. weaker model. Fixing the first layer of either DeepSeek-V3 or Llama3-8b, we observe a similar behavior to Fig. 11, namely the asymmetry in II is dominated by late layer against early layers, where the splitting appears roughly after the middle of the networks. Fixing the central layer of Llama3-8b (panel b)), we observe a clear splitting after the center of DeepSeek-V3, where the late layers predict the central layer of Llama3-8b better than the reverse. This effect is not observed fixing a central layer of DeepSeek-V3 in panel a), possibly because of an approximate compensation between the

fact that DeepSeek-V3 representations are in general more informative than Llama3-8b's, and the fact that the difference in depths induces a contrary asymmetry of comparable magnitude. A similar phenomenon appears when fixing the last layer of each model. In panel a), we see a large and stable gap, where the last representation of DeepSeek-V3 is more predictive than any layer from Llama3-8b. In contrast, when fixing the last layer of Llama3-8b, we observe an inversion roughly at the center of DeepSeek-V3, where the first half of the curve seems to be dominated by layer difference, and the second half by model quality and size.

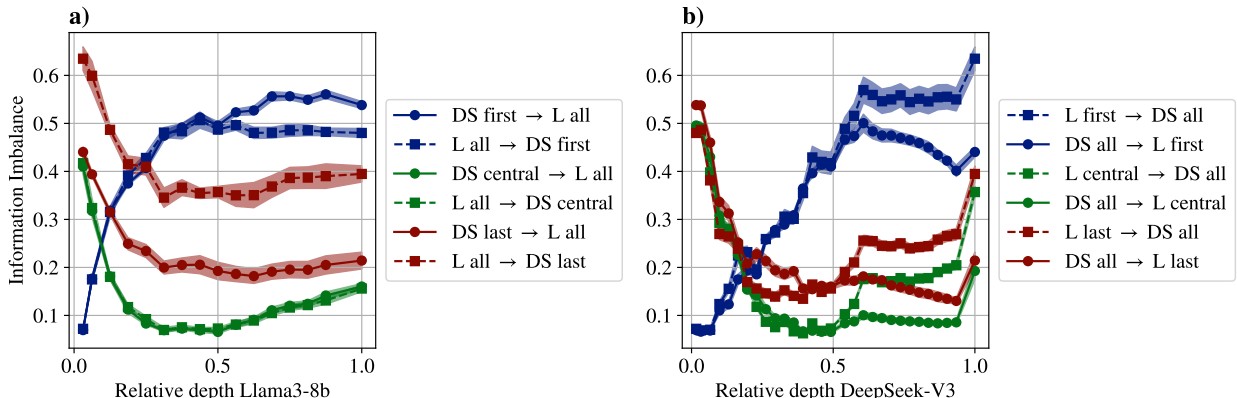

Figure 12: **Predictability between different layers, fixing language and changing model.** Information Imbalance between the representations on a fixed layer generated by one model and all layers of another, both processing the same English sentences, as a function of relative depth. In panel a), we fixed layers of DeepSeek-V3, abbreviated as DS in the legend. In panel b), we fixed layers of Llama3-8b, abbreviated as L in the legend. As representative layers of different processing stages, we picked the *first*, *central* and *last* layers of each model.

### F.3 Different language, different model, different layers

In Fig. 13, we have a set of activations produced by Llama3-8b processing English sentences and a set of activations produced by DeepSeek-V3 processing the corresponding Hungarian translations. Again, the central layers are the best predictors across the board. Contrary to what is seen in Fig. 11 and Fig. 12, and similar to the results in Fig. 2 and Fig. 3, there is very poor semantic predictability when the first layers are involved, most notably when fixing the initial layer encoding to Hungarian in panel a). Fixing a central layer of DeepSeek-V3's Hungarian representations in panel a) shows a broad II minimum at relative depths of Llama3-8b between roughly 0.25 and 0.5, whereas fixing a central layer of Llama3-8b's English representations in panel b) shows a broad II minimum at relative depths of DeepSeek-V3 between roughly 0.45 and 0.8, consistently with our results from Fig. 3 b). Finally, as observed in Figs. 11 and 12, the last layer representation shows once more to be more informative than earlier layers in both panels of Fig. 13.

## G Further language comparisons

In this Appendix, we report the Information Imbalance between English and several languages from different linguistic families, using parallel corpora from Opus Books (Tiedemann, 2012) and WMT17 (Bojar et al., 2017): Latvian, Czech, Dutch, Italian, Russian, French, German, Spanish, Chinese, Hungarian, Turkish and Finnish. Note that the latter 4 languages are not Indo-European, and that Russian and Chinese are written in non-Latin scripts. We used Qwen3-8b, as it offers a good tradeoff in terms of quality, size and multilingual support. We used 1,000 samples for each language, with token lengths between 30 and 80.

Fig. 14 shows that all curves follow a qualitatively similar layer profile, analogous to those we reported in Fig. 3 for a subset of languages and the larger DeepSeek-V3 model. Fig. 15 shows that better predictability

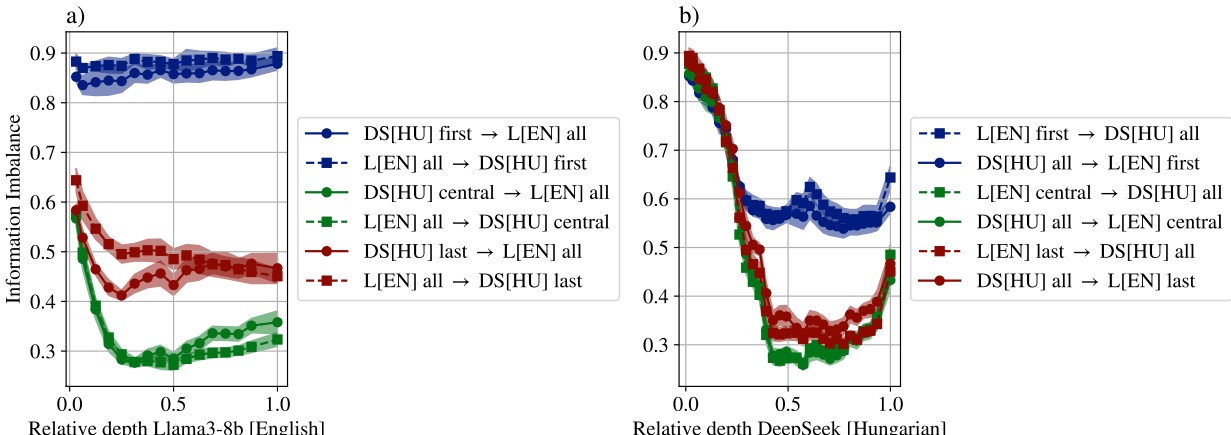

Figure 13: **Predictability between different layers, changing language and model.** Information Imbalance between the representation from a fixed layer generated by one model and all layers of another, as a function of relative depth. In panel a), we fixed layers of DeepSeek-V3 processing Hungarian sentences, abbreviated as DS[HU] in the legend. In panel b), we fixed layers of Llama3-8b processing the English translations, abbreviated as L[EN] in the legend. As representative layers of different processing stages, we picked the *first*, *central* and *last* layers of each model.

(lowest II) between English and the other languages correlates with the amount of content available for each of them on the Web.[2] The only outlier in the plot is Chinese. Removing it from the analysis gives a Pearson correlation coefficient of $r = -0.86$, with a $p$-value $p = 0.0006$, using the "PermutationMethod" from the scipy.stats (if we include Chinese, the correlation is at $r = -0.56$, with $p = 0.06$). We believe that the content estimate for Chinese in our source is unrealistically small. Qwen3-8b is, moreover, a model developed in China, which makes it very likely that it has been trained on large amounts of Chinese text. In general, though, our results tentatively support the conjecture that data availability is an important factor in determining convergence across language representations. The correlation with language family is more difficult to quantify with these data, since, excluding Chinese for the reasons discussed above, we have only three non Indo-European languages. Even if their II curves tend to be higher in Fig. 14, further experiments are required to reach robust conclusions.

## H Image captions

We report additional image–caption analyses that complement the main results of Sec. 3. Fig. 16 a) repeats the cross-modal experiment of Fig. 6 b) using LLama3.1-8B (fixed at ≈ 59% relative depth, in the minimum identified by Fig. 2) instead of DeepSeek-V3 as the text encoder. The qualitative picture is unchanged: DinoV2 achieves stronger cross-modal alignment than image-gpt-large, and the same information asymmetry in favour of the LLM is observed, confirming that the results are not specific to DeepSeek-V3.

Fig. 16 b) probes the effect of vision-model scale by comparing DeepSeek-V3 (fixed at ≈ 60% relative depth) against three DinoV2 variants (large, base, and small). All three profiles reach their minimum at the last layer, and larger models yield lower Information Imbalance, consistent with the Platonic representation hypothesis.

Fig. 16 c) verifies that the choice of fixed LLM layer does not qualitatively affect the results. We sweep all DeepSeek-V3 layers while fixing DinoV2 at its last layer and image-gpt-large at ≈ 42% relative depth.

---

[2]As reported in the "Usage statistics of content languages for websites" table of `https://en.wikipedia.org/wiki/Languages_used_on_the_Internet`.

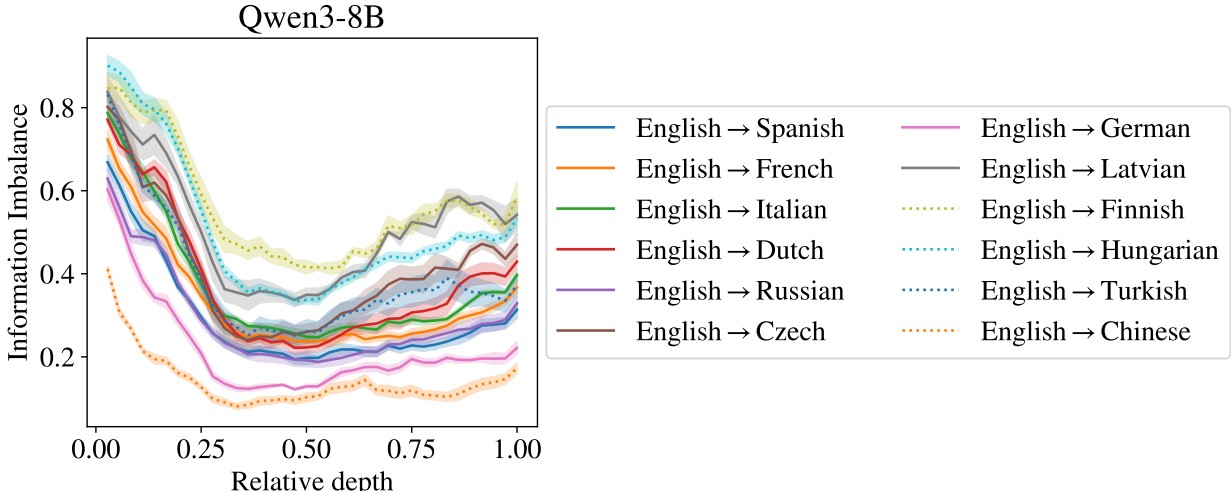

Figure 14: **More languages.** Information Imbalance from English to several languages, computed on representations generated by Qwen3-8b. The non-Indo-European languages are displayed with a dotted line, and the Indo-European languages are displayed with a solid line.

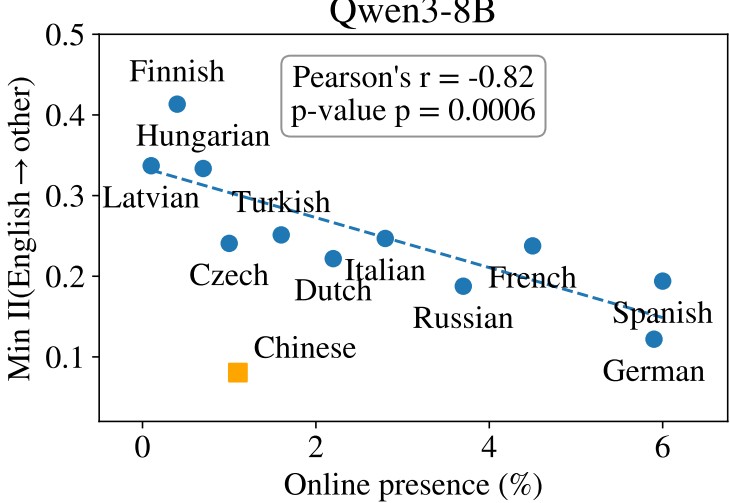

Figure 15: **Correlation between best semantic predictability and language online presence.** Minimum Information Imbalance from English to other languages (Min II(English → other)) as a function of "Usage statistics of content languages for websites" from `https://en.wikipedia.org/wiki/Languages_used_on_the_Internet` (Online presence (%)). The blue dashed line shows a linear fit for visual guidance only. Chinese is plotted with a different color and marker to highlight that it is the only outlier, and it does not enter in the linear fit, nor in the reported Pearson's $r$ in the figure. Including Chinese into the computation, we obtain $r = -0.56$ with $p = 0.06$

The Information Imbalance stabilizes after roughly 60% relative depth in DeepSeek-V3, confirming that the fixed-layer results reported in the main text are robust.

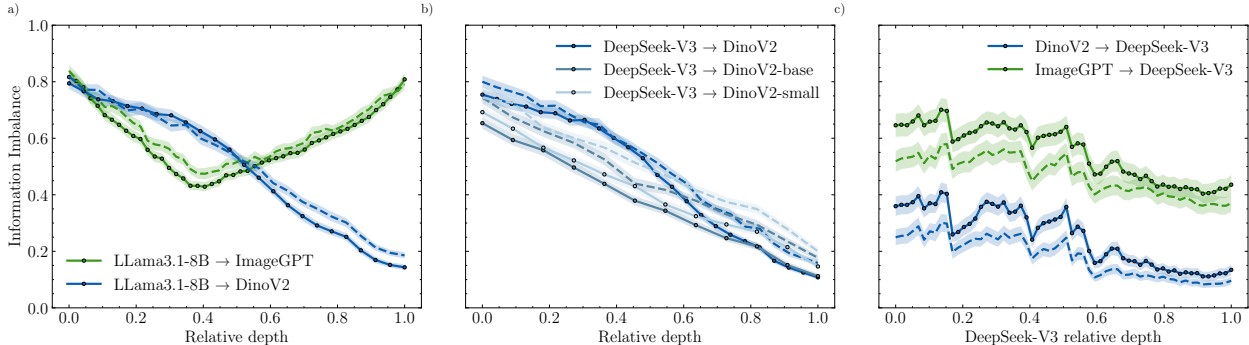

Figure 16: **a)** Cross-modal Information Imbalance on Flickr30k image–caption pairs with LLama3.1-8B ($\approx 59\%$ relative depth) as the text encoder, swept against DinoV2-large and image-gpt-large. **b)** DinoV2 model-size comparison: DeepSeek-V3 ($\approx 60\%$ relative depth) swept against DinoV2-large, DinoV2-base, and DinoV2-small. **c)** Information Imbalance from DinoV2 (last layer) and image-gpt-large ($\approx 42\%$ relative depth) against DeepSeek-V3. Solid lines show the Information Imbalance in the $A \rightarrow B$ direction; dashed lines show the reverse direction ($B \rightarrow A$), keeping the same model fixed at the same layer.

