# OpenReview forum: "A quantitative analysis of semantic information in deep representations of text and images"
_TMLR — Accepted by TMLR_

### Review · Reviewer_a4v2 · 2026-04-02

**Summary Of Contributions:**

The paper presents a measure called Information Imbalance (II) to measure the presence of semantic information in language and image models by measuring II across translations, images from same class and image caption pairs. They find that language models achieve the most semantic information in the middle layers, English representations are richer (due to more training data), image representations can have semantic information in middle layers or last layers depending on training methodology and large unimodal encoders contain more semantic information than joint embeddings. They also find averaging tokens achieves highest semantic information compared to concatenating tokens or using the last token.

Strengths:
- The paper provides extensive evaluations on translations, image pairs and image caption pairs using different LLMs and also different scales.
- The observations are well articulated and possible interpretations are provided.

Weaknesses:
- The rationale behind the requirement of Information Imbalance (II) is not well explained. The formula of II is also not well defined.
- The authors claim: "Namely, if one measures an II between two representations of, say, 0.2, it can be interpreted as them sharing roughly 25 percent of their features", but they do not explain how they arrive at this conclusion as II measures rank of neighbors, how does it get to shared features?
- Most of the experiments are conducted using English and Italian, it would be useful to show another language like Hungarian.
- The work focuses on European languages, it will be interesting to see how languages Mandarin or Hindi will interact in this framework. Also the work does not analyse very well if similar language origins and structure result in better II. Like Italian and French compared to Italian and English. It only compares all to English.
- The paper does not provide rationale behind why CLIP performs better than ImageNet-Deepseek V3. And the claims regarding joint embedding models performing worse than large unimodal models is somewhat unfounded and less concrete.
- Also since the paper focuses on cross language or modality semantic overlaps, it misses an significant issue, maybe layer 5 has the most semantic information for Hungarian while Layer 10 has it for English. They only compare across same layers.

**Audience:**

Yes

**Audience Explanation:**

The work provides an interesting study on the interpretability of text and image representations, finding how much information is carried across in translations, image pairs of same class and image caption pairs.

**Claims And Evidence:**

No

**Claims Explanation:**

The work conducts experiments showing results which back most of their claims. However I have three issues:

1) The authors claim that Information Imbalance is needed for this analysis. However, it is not clear how Neighborhood overlap is worse than II.
2) The authors claim large unimodal models are better for cross-modal representations over small joint embedding models like CLIP. However CIP performs much better than Image-Net/Deepseek-V3 and no explanation is provided on this.
3) Why do the authors only compare the same layers for different languages? Why not also compare layer 5 of Hungarian with layer 10 of English?

**Requested Changes:**

- Provide a rationale behind the requirement of Information Imbalance (II) over Neighbor Overlap.
- The authors write in text that j is the nearest neighbor of i in X. However in the formula it seems like they iterate over all possible i,j s in which case all average ranks will be the same as we will find ranks 1 to N going through all js. This needs to be clarified.
- The authors need to back the claim: "Namely, if one measures an II between two representations of, say, 0.2, it can be interpreted as them sharing roughly 25 percent of their features". How do they arrive at this conclusion as II measures rank of neighbors, how does it get to shared features?
- Most of the experiments are conducted using English and Italian, it would be useful to show another language like Hungarian.
- The work should include other languages like Mandarin and Hindi and analyse if having similar origins results in better semantic overlaps or if the overlaps happen at different regions in the model.
- The paper should provide rationale behind why CLIP performs better than ImageNet-Deepseek V3. And the claims regarding joint embedding models performing worse than large unimodal models is somewhat unfounded and less concrete.
- Provide rationale of why they only compare the same layers. Different languages or modalities might contain semantic information at different layers. Like layer 10 of English should be compared to Layer 5 of Hungarian.

---

> ### Author Response · Authors · 2026-05-08
>
> Thanks for the constructive feedback. Changes are in red in the revision.
>
> Requested changes:
>
> ## 1:
> The Information Imbalance (II) is related with the Neighborhood Overlap (NO), given that both compare representation neighborhoods. The main advantage of II lies in its asymmetric nature, as we know emphasize in the introduction. The asymmetry allows one to identify which of two representations is more informative. If one uses a symmetric metric and gets a poor alignment value between two models, one cannot tell whether this is because the two representations are different or because one of them is of poor quality or not sufficiently informative.
>
> ## 2:
> The expression $\sum_{i,j:r_{ij}^X = 1}$ is a constrained sum: it ranges over pairs $(i, j)$ such that $j$ is the nearest neighbour of $i$ is $X$ (i.e., $r_{ij}^X = 1$). For each $i$, there is exactly one such $j$, so the sum has $N$ terms, not $N(N-1)$. We have clarified this.
>
> ## 3:
> For Fig. 1-b), we constructed a toy model of two datasets explicitly sharing a given percentage of their features, finding that the fraction of shared features is roughly 0.2, when II≈0.25. This doesn’t mean a given value of II must always be produced by data with this structure. E.g., Fig 1-a) illustrates a setup in which a similar value is observed in different conditions. We clarified this in section 2.3.
>
>
> ## 4 & 5:
> To be systematic, when changing a parameter such as token number (Fig. 2-a) and 2-b)) or LM (Fig 3-b), we kept the second language fixed to Italian. For a fair comparison, Figs. 4-a) and 5 also fix the languages to English and Italian. Nonetheless, Figs 3-a) and 4-b) show that three other languages behave like Italian. The only exception is Hungarian (Fig 3-a), which, as stated in Sec. 3.1.2, is a language with almost one order of magnitude less resources on the internet. We have added in Appendix F a more extended analysis with several new languages from multiple language families. We confirm a strong correlation between available resources and representational convergence as measured by II.
>
> ## 6:
> In Fig. 6b, the minimum II values are ≈0.10 for DeepSeek-V3+DinoV2, 0.25 for CLIP, and 0.40 for DeepSeekV3+image-gpt-large. CLIP thus outperforms DeepSeek-V3+ image-gpt-large, but underperforms DeepSeek-V3+DinoV2:  the jointly trained models outperform one unimodal pair. The contrast between DinoV2 and image-gpt-large implies that despite being smaller (300M vs 1.4B, respectively), DinoV2 groups same-class images more tightly (Fig. 6a, min II ≈ 0.35 vs. min II ≈ 0.65), and this gap propagates to the cross-modal pairing in Fig. 6b. The relevant factor is therefore not encoder size but how each training objective shapes the resulting representations. The CLIP comparison reinforces this from a different perspective: small but contrastively-trained encoders (123M text, 300M image) outperform the much larger autoregressive pair, while still falling short of DeepSeek-V3+DinoV2, perhaps due to the substantial scale gap on the language side. This is consistent with [1], who find that size and model quality rival the performance of multimodal alignment in zero-shot tasks. We have revised §3.2.2 to improve clarity.
>
> ## 7:
> In Sec. 3.1, we presented our results constrained to equal-depth comparisons, to simplify the presentation and visualization. In the new Appendix E we study II between different layers. We look at same-model layers processing the same sentences, different models processing the same sentences, and different models, each processing a different language.
> We find that later layers typically predict earlier layers better than the reverse. The cross-layer comparisons also confirm that the central layers are particularly information-rich, displaying the highest predictability scores.
> This information asymmetry between layers is interesting and relevant, but  can be misleading when measuring the relative information content between representations, and, in order to make the message more clear, we decided to focus on similar-depth representations in the main study. We have clarified this at the end of section 3.1.1.
>
> [1] https://arxiv.org/abs/2210.01738
>
> [2] https://arxiv.org/abs/1807.03748
>
> [3] https://www.pnas.org/doi/10.1073/pnas.2317256121

---

### Review · Reviewer_5f6W · 2026-04-19

**Summary Of Contributions:**

This paper provides a quantitative analysis of how semantic information is organized in deep representations across languages, models, and modalities, using Information Imbalance as a metric of directed predictability between representations rather than standard symmetric similarity measures. It shows that semantically related inputs, such as translations, same-class images, and image-caption pairs, tend to align most strongly in specific layers rather than uniformly across the network. The numerical resuls also find that sentence semantics are distributed across multiple tokens rather than concentrated only in the last token, and that representation alignment is often asymmetric, with English and larger models being more predictive of other languages or smaller models than the reverse.

Strengths

- 1. It studies representation alignment with a genuinely directional metric rather than only symmetric similarity measures. By using Information Imbalance, the paper can distinguish whether one representation is more predictive of another, which allows it to make more nuanced observations than CKA or Neighborhood Overlap.

- 2. **The breadth of the empirical study across both text and vision**. The paper does not restrict itself to one dataset or one model family: it analyzes translation pairs from Opus Books using DeepSeek-V3 and Llama3 (1B/3B/8B), same-class image pairs from ImageNet-1k using image-gpt-large and DINOv2-large, and image-caption pairs from Flickr30k using DeepSeek-V3, DINOv2, image-gpt, and CLIP.

Weaknesses: The main weakness is that many of the paper’s conclusions remain correlational and somewhat over-interpreted. For instance, the finding that DeepSeek-V3 + DINOv2 exhibits lower Information Imbalance than the jointly trained CLIP pair is interesting, but the paper goes further to suggest that model scale may outweigh explicit multimodal training. That interpretation feels too strong, because the comparison mixes several confounding factors at once, including model size, architecture, training objective, and representation type. Similarly, the claim that English representations are more informative because English is a higher-resource language is plausible, but it is not directly tested.

**Additional Comments:**

N.A.

**Audience:**

Yes

**Audience Explanation:**

The paper would likely be of interest to readers, since it studies a broad and timely question: how semantic information is distributed across layers and whether semantically related inputs from different languages or modalities converge to similar internal structures. Its findings on middle-layer semantic alignment in translation models, the comparison of DeepSeek-V3 and Llama3, and the cross-modal analysis involving DINOv2, image-gpt, and CLIP give it relevance beyond a single benchmark or architecture.

**Broader Impact Concerns:**

N.A.

**Claims And Evidence:**

Yes

**Claims Explanation:**

The layerwise Information Imbalance results consistently support the paper’s central observations, such as strongest semantic alignment in middle layers for autoregressive models and later layers for encoder-style vision models, as well as asymmetries between languages and model sizes.

**Requested Changes:**

Refer to the weaknesses in the summary, my requested changes are mostly aimed at improving the clarity of the claims.  1) The authors should tone down or better justify some of the broader causal interpretations, especially the suggestion that model scale may outweigh explicit multimodal training based on the DeepSeek-V3 + DINOv2 vs. CLIP comparison, 2) and the claim that English is more informative because it is a higher-resource language; these explanations are plausible, but the current evidence is more correlational than decisive.

---

> ### Author Response · Authors · 2026-05-08
>
> We thank the referee for their positive evaluation of our manuscript.
>
> ## Requested Change 1)
> We will revise paragraph §3.2.2 in order to avoid overstatements. We find that across our configurations (three vision encoders, three text encoders, three DinoV2 scales), three observations stand out. (i) Scale alone is not sufficient. Image-gpt-large (1.4B), the largest vision encoder we consider, paired with DeepSeek-V3 yields the highest II of the three pairs in Fig. 6b, worse than both DeepSeekV3 + DinoV2 and CLIP. The same gap is visible within-modality in Fig. 6a (min II ≈ 0.65 for image-gpt-large vs min II ≈ 0.35 for DinoV2). (ii) Scale does matter once representation quality is held fixed. Appendix G (Fig. 15b) shows that DeepSeek-V3 paired with DinoV2 large/base/small gives 0.10/0.10/0.15, respectively. (iii) Joint training is not necessary, but it is effective. CLIP (123M text, 300M image) reaches a minimum II than the much larger DeepSeek-V3 + image-gpt-large pair, while still falling short of DeepSeek-V3 + DinoV2. This is consistent with what also reported by Norelli at el. (2023) on size/model quality making up for lack of multimodal alignment during training.
>
> ## Requested Change 2)
> We agree with the referee on the fact that our evidence on this matter is correlational and not causal, and we were careful in our phrasing in sections 3.1.2 and 3.1.3 to avoid overclaims. Nonetheless we have clarified this, and we added an Appendix F with the Information Imbalance between English and several languages to explore this issue in more detail. The list of languages in this Appendix is Latvian, Czech, Hungarian, Dutch,  Italian, Russian, French, German, Turkish, Finnish, Chinese, and Spanish. The predictability between English and other languages correlates indeed with the online presence of each language. The only outlier seems to be Chinese, for which we observed that using Qwen3-8b, a Chinese model, English-Chinese presents the lowest II among all the studied languages.  For all languages in our list excluding Chinese, we report a correlation with Pearson’s r= -0.82 and p-value p=0.0006 between the “Usage statistics of content languages for websites“ from https://en.wikipedia.org/wiki/Languages_used_on_the_Internet and the minimum II across layers between English and each language, noting, however, that the online presence of each language is only a proxy of their presence in the training set of an LLM. Including Chinese, we obtain Pearson’s r= -0.56 and p-value p=0.06

---

### Review · Reviewer_3yju · 2026-04-30

**Summary Of Contributions:**

This paper presents a study of the similarity of representation of different model sizes (DeepSeek and LLama), different modalities (text DeepSeek and vision Dino) and different domains (different languages for text). First, the paper argues that the previous distances are not adapted in certain cases and proposes to use the Information Imbalance, which is a proxy of the cross-entropy, and it is asymmetric. With this tool, the authors study the representation layers that are more aligned, the encoding of the textual representation that works better, the differences in model sizes, different languages for text, different vision models, and finally, cross-modal for images and their captions.

**Audience:**

Yes

**Audience Explanation:**

I think the Platonic hypothesis is very interesting and can lead to new ways to train our models. However, as I said above, I would like to start to see not only analysis but also development of methods based on this hypothesis. This would provide more concrete evidence of the effectiveness of the hypothesis.

**Broader Impact Concerns:**

No concerns.

**Claims And Evidence:**

No

**Claims Explanation:**

The proposed study is quite interesting; however, the motivation for using Information Imbalance as a measure of alignment of representations is not very convincing.
- The fact that the measure is asymmetric is not an advantage by itself. I could not find why an asymmetric measure is better than a symmetric.
- The evaluation on a synthetic dataset is limited. There are other measures that should be considered, for instance Canonical Correlation Analysis, and its variants, Representational Similarity Analysis and mutual k-Nearest Neighbors, among others.
- Looking at fig. 1, the authors provide some justification for preferring II, but those justifications are not very convincing. For instance, they mention that in fig. 1(c): CKA saturates early and is not very discriminative; however, the CKA curve is very similar to the II.
- The authors mention that Information Imbalance is a proxy of cross-entropy; however, no explanation of this is given.

The experiment showing that the II of clip text and vision is lower than that of models trained on one of the two modalities independently is not very convincing. How is it possible that modalities trained independently are better aligned than modalities trained to be explicitly aligned? Can it be due to the fact that Clip text encoder can store only a limited number of tokens, and therefore this is not enough for the captioning on Flicker 30k?

Similar to the previous paper, this contribution is mostly an analysis of the similarity of different representations following the Platonic hypothesis. However, in my opinion, it is important to go beyond the analysis and show the practical implications of this hypothesis. For instance, is it possible to align different modalities without paired data? Is the degree of alignment enough for such kind of applications?

**Requested Changes:**

- (critical) Answer to my points about Fig. 1, and find better ways to justify and support the use of II as a measure for the alignment.
- (critical) Analyse more in detail the experiment of Fig. 6b. In my opinion, there should be a clear reason why the alignment of Clip text and vision is worse than separated modalities. The size of the model is not really convincing to me. Can it be that the proposed measure is very different than the InfoNCE used in Clip?
- (not critical) Illustrate the proposed II, to get a better grasp of how it works.
- (not critical) Find possible ways to use the proposed analysis for tangible applications.

---

> ### Author Response · Authors · 2026-05-08
>
> Thanks for your feedback. Changes are shown in red in the revision.
>
> ## Requested change 1)
> The asymmetry of II allows one to identify which of two representations is more informative. If one uses a symmetric metric and gets a poor alignment value between two models, one cannot tell whether this is because the two representations are different or because one of them is of poor quality or not sufficiently informative. We have clarified this in the introduction.
> The two synthetic Gaussian datasets in Fig. 1 provide simple setups to interpret and capture information asymmetries. They are not meant as extended benchmarks.
>
> As mentioned in Sec. 2.3, Neighborhood Overlap (NO) was already found to be preferable to CKA by [2] (see their Appendix A). [4] showed moreover that local metrics based on neighborhoods are more robust than global metrics based on distances, such as CKA. The II, already successfully used in [5], is a local metric based on neighborhoods, very similar to NO, with the additional feature of being asymmetric.
>
> About “The evaluation on a synthetic dataset is limited. [...]”, we highlight that
> CKA was explicitly designed by [3] to overcome limitations observed in CCA: we thus did not experiment with CCA.
> Mutual k-Nearest Neighbors is our Neighborhood Overlap, where we follow the naming in [2].
> We added Representation Similarity Analysis to Figure 1, showcasing that its behaviour is qualitatively similar to NO.
> Regarding “The authors mention that Information Imbalance is a proxy of cross-entropy; however, no explanation of this is given.” We now describe in Appendix A the connection between II and information theory established by [6].
>
> [1] https://arxiv.org/pdf/2106.07682
>
> [2] https://arxiv.org/abs/2405.07987
>
> [3] https://arxiv.org/abs/1905.00414
>
> [4] https://arxiv.org/pdf/2602.14486
>
> [5] https://openreview.net/forum?id=0fD3iIBhlV
>
> [6] https://www.pnas.org/doi/full/10.1073/pnas.2317256121
>
> ## Requested change 2:
> Re-examining Fig. 6b, the minimum IIs are 0.10 for DeepSeek-V3+DinoV2, 0.25 for the CLIP, and 0.40 for DeepSeek-V3 + image-gpt-large. Image-gpt-large (1.4B parameters) is in fact the largest vision encoder we consider, exceeding both DinoV2 and the CLIP visual encoder (≈ 300M each), yet yields the worst alignment. This rules out scale as the only responsible for the effect. The worse alignment of CLIP is instead consistent with the within-modality semantic alignment of each vision encoder in Fig. 6a (DinoV2 reaches ≈ 0.35, image-gpt-large plateaus at ≈ 0.65). CLIP’s intermediate position also shows the effect of the training objective, as its joint contrastive training compensates substantially for its modest scale (text: 123M, image: 300M), even though it does not bridge the gap to the much larger DeepSeek-V3 + DinoV2 pair. This is consistent with Norelli et al. (2023), who align frozen unimodal encoders to match CLIP-level zero-shot performance. Finally, to investigate  whether the effect can be due to CLIP's token window, we recomputed the II on the subset of images whose 5-caption concatenation fits within CLIP's 77-token context. The ordering is unchanged: at the last layer, DeepSeek-V3 + DinoV2 still attains the lowest value (~0.11) while CLIP reaches ~0.18.
>
> Regarding the InfoNCE / II relationship: the InfoNCE loss was introduced by Oord et al. (2018) as a lower bound on mutual information that tightens with the number of contrastive samples; minimising it is equivalent to maximising the lower bound on $I(X; Y)$, hence to minimising conditional cross-entropy. The Information Imbalance is a rank-based proxy for the restricted mutual information (Del Tatto et al., 2024), which is a similar quantity. As such, there are no obvious confounding issues due to the two measures.
>
> ## (not critical 1)
>
> We added a sentence in the introduction stressing that a simple visual example of the metric can be found in the reference that first defined it.
>
> ## (not critical 2)
>
> Possible applications include deciding the optimal layers to align data across different modalities (e.g., for cross-modal retrieval) without the need to train a multimodal model, as well as applications in model stitching to share pre-trained resources (e.g., applying a classifier trained on modelA to representations produced from modelB). Note that the asymmetric property of Information Imbalance makes it particularly well-suited for such applications because it tells us which of two models or input categories contains more information than the other: for example, if modelA can predict modelB’s representations but not vice versa, it will make sense to transfer information from modelA to modelB, but not the opposite.

---

### Author Response · Authors · 2026-06-25
**Official Comment**

We thank the Action Editor and the reviewers for the positive recommendation. We have uploaded a revised version that addresses the three outstanding points:
1. **Causal interpretations.** We have softened the causal phrasing and now present the relevant analyses with explicit caveats, stating that the evidence on model scale vs. multimodal training and on English being a higher-resource language is correlational, not causal. We have reviewed the manuscript, as per the reviews, and reviewed the conclusions.
1. **Figure 1.** To improve clarity and focus on the statistical power and the asymmetry of the information imbalance, we have moved the very-high-dimensional panel (former Fig. 1c, p=10^5) and its discussion to a dedicated appendix. Figure 1 now keeps only panel a) (recovering the direction of predictability) and panel b) (p=10^2, where the same asymmetry is visible in the gap between the two II curves).
1. **Applications of the asymmetry.** We have expanded the discussion in the conclusions, noting how the asymmetry of the Information Imbalance can be leveraged to perform feature selection in the representation space when the information content of two representations is
asymmetric. This would allow one to detect which features are responsible for the symmetry breaking: a relevant step further in the analysis of how LLMs encode meaning.

---

### Decision · Action_Editor_KLmt · 2026-06-20

**Recommendation:** Accept with minor revision

**Additional Comments:**

As mentioned above, with the positive recommendation, there are still some outstanding concerns from the reviewers. The paper needs a minor revision to clear these issues before the final Accept. These concerns include:

- Clearly tone down the causal interpretations and present the new analysis with appropriate caveats.

- Revision of Fig.1 to better convey the motivation of the paper. *While Fig.1(a) shows that information imbalance can extract information about the relative importance of a modality, Fig.1(b) and (c) show the problem of CKA, which is not related to the proposed contribution and already shown in previous work.*

- It would be great to clarify more about the potential applications of the asymmetry of the proposed measure, although testing one of the applications (asked by the reviewer) may be too much beyond a minor revision.

**Audience:**

Yes

**Audience Explanation:**

Given the interesting and broad question studied in this work, there will be a group of audience in TMLR interested in knowing the findings of this paper, and potentially inspire new follow-up studies.

**Claims And Evidence:**

Yes

**Claims Explanation:**

This paper was reviewed by three experts with detailed comments. In the first round of review, there were concerns regarding the unclear/unconvincing motivation, unconvincing experiments, practical applications, overclaims, and some missing details. After the authors' responses to the concerns, most of them were addressed, as acknowledged by the reviewers, with some minor issues outstanding (see below for details).

That said, the major issues were cleared and the claims made in the submission were supported by evidence. All reviewers recommended accept (with 2 *Leaning Accept* and 1 *Accept*). Given the contributions made by this paper and the agreed strengths around the representation alignment and the extensive empirical analysis, the AE would recommend Accept, if the remaining minor issues are cleared in the minor revision.